# Mild Cognitive Impairment Is Associated with Poorer Nutritional Status on Hospital Admission and after Discharge in Acutely Hospitalized Older Patients

**DOI:** 10.3390/geriatrics7050095

**Published:** 2022-09-10

**Authors:** Olivia Bornæs, Aino L. Andersen, Morten B. Houlind, Thomas Kallemose, Juliette Tavenier, Anissa Aharaz, Rikke L. Nielsen, Lillian M. Jørgensen, Anne M. Beck, Ove Andersen, Janne Petersen, Mette M. Pedersen

**Affiliations:** 1Department of Clinical Research, Copenhagen University Hospital Amager and Hvidovre, 2650 Hvidovre, Denmark; 2The Hospital Pharmacy, The Capital Region of Denmark, Marielundsvej 25, 2730 Herlev, Denmark; 3Department of Drug Design and Pharmacology, University of Copenhagen, Universitetsparken 2, 2100 Copenhagen, Denmark; 4The Emergency Department, Copenhagen University Hospital Amager and Hvidovre, Kettegaard Allé 30, 2650 Hvidovre, Denmark; 5Dietetic and Nutritional Research Unit, Copenhagen University Hospital Herlev-Gentofte, Borgmester Ib Juuls Vej 50, 2730 Herlev, Denmark; 6Department of Clinical Medicine, Faculty of Health and Medical Sciences, University of Copenhagen, Blegdamsvej 3B, 2200 Copenhagen, Denmark; 7Copenhagen Phase IV Unit, Center of Clinical Research and Prevention, Department of Clinical Pharmacology, Copenhagen University Hospital Bispebjerg and Frederiksberg, Nordre Fasanvej 57, 2000 Frederiksberg, Denmark; 8Section of Biostatistics, Department of Public Health, University of Copenhagen, Øster Farimagsgade 5, 1014 Copenhagen, Denmark

**Keywords:** malnutrition, nutritional status, cognitive dysfunction, acute admission, hospital, frailty, comorbidity, older adults, medication

## Abstract

In acutely hospitalized older patients (≥65 years), the association between mild cognitive impairment (MCI) and malnutrition is poorly described. We hypothesized that (1) MCI is associated with nutritional status on admission and after discharge; (2) MCI is associated with a change in nutritional status; and (3) a potential association is partly explained by frailty, comorbidity, medication use, and age. We combined data from a randomized controlled trial (control group data) and a prospective cohort study (ClinicalTrials.gov: NCT01964482 and NCT03052192). Nutritional status was assessed on admission and follow-up using the Mini Nutritional Assessment—Short Form. MCI or intact cognition (noMCI) was classified by three cognitive performance tests at follow-up. Data on frailty, comorbidity, medication use, and age were drawn from patient journals. MCI (*n* = 42) compared to noMCI (*n* = 47) was associated with poorer nutritional status with an average difference of −1.29 points (CI: −2.30; −0.28) on admission and −1.64 points (CI: −2.57; −0.70) at 4-week follow-up. Only age influenced the estimates of −0.85 (CI: −1.86; 0.17) and −1.29 (CI: −2.25; −0.34), respectively. In acutely hospitalized older patients, there is an association between MCI and poorer nutritional status upon admission and four weeks after discharge. The association is partly explained by higher age.

## 1. Introduction

Worldwide, the population of older adults (≥65 years) is increasing [1]. In 2018, in Denmark, older adults constituted 45% of all acute hospital admissions and this proportion is expected to increase [2]. Both malnutrition and cognitive impairment are prevalent in persons aged ≥65 years, who are admitted to emergency departments (ED) [3,4,5,6,7,8,9]. Of these patients, 60–70% are malnourished or at risk of malnutrition on admission and 27–40% are cognitively impaired [3,4,5,6,7,8,9]. Further, the prevalence of mild cognitive impairment (MCI) is unknown in this population. Today, guidelines on nutrition in dementia exist, but these do not include patients with MCI [10]. Thus, no targeted treatment for malnutrition exists in older adults with MCI.

Both malnutrition and MCI are associated with readmission, mortality, and functional decline one year after hospitalization [4,6,9,11]. Even so, few studies have investigated the association between malnutrition and MCI [12,13,14]. Three studies have found a higher frequency of malnutrition or risk of malnutrition in institutionalized or hospitalized older adults with MCI compared to those with normal cognitive function for age [12,13,14]. Further, Khater et al. found a higher frequency of MCI in those who were malnourished or at risk of malnutrition compared to those who were well-nourished. The authors also found that a nutritional deficit and MCI were strongly associated after adjusting for potential confounders [14]. None of the studies, however, explored the association in acutely hospitalized older patients (≥65 years).

The presence of MCI in patients with malnutrition might explain the lack of effect of nutritional interventions that have been reported in recent studies in adults ≥65 with malnutrition during and after acute hospitalization [15,16,17]. Since persons with MCI have slight impairment of cognitive functions within memory and processing speed [18], they may lack the ability to understand and comply with dietary counseling.

Moreover, frailty, comorbidity, and medication use have been linked to cognitive decline and/or nutritional status. A systematic review by Brigola et al. reports an association between frailty and cognitive function, especially memory [19], with a higher prevalence of cognitive impairment in persons with frailty, and with a higher risk of MCI with greater frailty [19]. Moreover, risk factors for malnutrition are associated with frailty in older adults [15]. Further, in persons ≥65 years, multimorbidity is common and is associated with subjective cognitive decline and malnutrition [20,21,22]. Furthermore, in a study on patients who were acutely admitted to a medical department, half (53%) of those aged ≥65 years received hypnotic-sedative medication [23]. The use of medication with sedative properties or sedative-hypnotic medications has been linked to cognitive decline [23,24,25] and might impact the patient’s cognitive abilities, and as a result, this affects nutritional intake and, thus, the nutritional status.

Therefore, our hypotheses are (1) that MCI is associated with poorer nutritional status on admission and after discharge in acutely hospitalized older patients; (2) that MCI is associated with a change in nutritional status from admission to 4 weeks after discharge in acutely hospitalized older patients; and (3) that a potential association between MCI and nutritional status is partly explained by age, frailty, comorbidity, and use of sedative medication.

## 2. Materials and Methods

### 2.1. Study Design

This study is a prospective observational cohort study using data from two studies: STAND-Cph, a randomized controlled trial registered at Clinical Trials.gov (NCT01964482); and FAM-Cph, a prospective cohort study registered at Clinical Trials.gov (NCT03052192). Data from the STAND-Cph control group and the FAM-Cph cohort collected at baseline and four weeks after discharge (4-week follow-up) were aggregated. Reporting of this study adheres to the ‘STrengthening the Reporting of OBservational studies in Epidemiology (STROBE) guidelines’ using the checklist for cohort studies [26].

### 2.2. Setting

Data from STAND-Cph were collected between September 2013 and September 2018 and data from FAM-Cph were collected between November 2016 and August 2017. Participants in both STAND-Cph [27] and FAM-Cph were recruited at the Emergency Department (ED) of Copenhagen University Hospital, Hvidovre, Denmark, which counts approximately 12,600 acute admissions annually [28]. Baseline data were collected in the ED shortly after admission and 4-week follow-up visits were conducted by a research assistant in the participant’s own home.

### 2.3. Participants and Recruitment

Eligible participants were identified through the electronic patient journal. Participants in STAND-Cph and FAM-Cph were eligible if they were aged ≥65 years and admitted to the ED for an acute medical illness, and further, in FAM-Cph, were Caucasian. In STAND-Cph and FAM-Cph, eligible patients were excluded if they were: unable to understand and speak Danish, unable to cooperate, terminally ill or in isolation room stay, and further, in STAND-Cph, were transferred to another hospital or the intensive care unit, receiving treatment for cancer or unable to stand (for further details please see Pedersen et al [27] and Andersen et al. [29]). All participants in STAND-Cph were randomized to either a strength training intervention or standard care. The current study only included patients if they received standard care. All participants gave written informed consent before participation. All methods used for recruitment and follow-up were approved by the National Committee on Health Research Ethics in Denmark (STAND-Cph: H-2-2012-115, FAM-Cph: H-16038786). The data aggregation was approved by the Danish National Committee on Health Research Ethics on 12th April 2021 (Notification no. 78871). Permission to collect, store, and process data was obtained by The Danish Data Protection Agency (STAND-Cph: 2007-58-0015, FAM-CPH: 2012-58-0004).

### 2.4. Variables

Data collected at baseline (shortly after admission) and 4-week follow-up included descriptive variables and nutritional status. Further, at a 4-week follow-up, cognitive performance tests were completed and the results were used to characterize participants with or without MCI and further with MCI-subtypes within memory domains and/or non-memory domains [30]. Data on cognitive function were collected at the 4-week follow-up to ensure that the patient’s cognitive performance was not affected by the acute illness.

#### 2.4.1. Dependent Variable—Nutritional Status

The Mini Nutritional Assessment—Short Form (MNA-SF) was used for the assessment of nutritional status at baseline and 4-week follow-up. MNA-SF is a nutritional screening tool that has been validated to identify patients ≥65 years in an acute setting who are at risk of malnutrition or malnourished [31]. MNA-SF consists of six items: decline in food intake, weight loss, and acute disease/psychological stress within the past three months; and decline in mobility, neuropsychological problems, and Body Mass Index (BMI). MNA-SF scores range from 0 to 14, where scores ranging from 0 to 7 indicate malnutrition, 8 to 11 indicate risk of malnutrition, and 12 to 14 indicate a normal nutritional status [31]. MNA-SF was grouped as malnourished (MNA-SF score = 0–7), at risk of malnutrition (MNA-SF score = 8–11), or normal nutritional status (MNA-SF score = 12–14) when used for a descriptive purpose. For further analyses, the specific MNA-SF score was used. A difference in the MNA-SF score of ≥1 between MCI and no MCI was chosen as a clinically relevant difference, since a change of ≥1 is the minimal change needed to change nutritional status in consideration of the grouping of nutritional status in the MNA-SF [32].

#### 2.4.2. Independent Variable—Mild Cognitive Impairment

Mild Cognitive Impairment (MCI) is a condition where individuals show cognitive impairment greater than expected for age and educational level, with minimal impairment of one’s ability to perform activities of daily living [33,34]. In this study, the diagnostic criteria for MCI are based on performance on the following three tests at the 4-week follow-up:

Hopkins Verbal Learning Test, Revised (HVLT-R) is a valid instrument for clinical and research-based neuropsychological assessment of older patients [35]. HVLT-R is a learning and memory test, which consists of three learning trials, a delayed recall trial (20–25 min later) followed by a recognition task. At each learning trial, a target list of 12 words (categorical nouns) is read to the subject, who is asked to recall the words and repeat as many as possible after the read-aloud [36]. For the delayed recall trial, the subject is asked to repeat as many of the 12 words as possible. During the recognition task, the subject is asked to identify words from the learning trials, by responding “yes” or “no” to a list of 24 words (12 from the target list and 12 non-target words) [36]. Scores derived from HLVT-R are total recall (the sum of words recalled during trials 1–3), delayed recall (words recalled after 20–25 min), and recognition (the number of true-positive answers minus the number of false-positive answers from the recognition task). Higher scores indicate better performance.

The Symbol Digit Modalities Test (SDMT) is commonly used to measure attention [37]. Good performance on the SDMT requires a range of cognitive operations, such as memory, perceptual/processing speed, mental flexibility, and visual scanning [37]. The SDMT is presented on a single sheet of paper and consists of a series of symbols. Using a key matching nine symbols with numbers located on the top of the page, subjects are asked to consecutively match as many symbols with numbers as possible within 90 s [37]. The number of correct matches is recorded, with higher scores indicating better performance.

The Trail Making Test (TMT) is a neuropsychological instrument for detecting neurological disease and neuropsychological impairment [38]. The TMT measures processing speed, sequencing, mental flexibility, and visual-motor skills [38]. The TMT is performed on paper using a pen and consists of two parts, A and B. In part A, subjects are asked to connect 25 numbers in numerical order, and in part B, subjects are asked to connect 25 numbers and letters in numerical and alphabetical order, alternating between numbers and letters [38]. The time taken to complete parts A and B was recorded, with 300 s as the maximum score. Faster completion indicates better performance.

Performance on HVLT-R total recall, HVLT-R delayed recall, HVLT-R recognition, DSST, TMT part A, and TMT part B were used to diagnose MCI. The scores from each test were converted into z-scores adjusted for age, based on normative data from healthy peers [36,39,40,41]. Using z-scores instead of the test score allows for comparison across age groups and cognitive measures [34]. A standard deviation of 1.5 was used as a cut-off value to identify impaired performance on a test, which is consistent with other studies [34,42]. If patients had z-scores below 1.5 in ≥2 of 6 tests they were identified with MCI and if not, they were considered cognitively intact (noMCI). Three different subtypes of MCI were identified: single domain amnestic MCI (aMCI) if participants had z-scores < 1.5 in ≥2/6 tests within the memory domain; single domain non-amnestic MCI (naMCI) if participants had z-scores < 1.5 in 2/6 tests within the nonmemory domain; and multiple domain MCI (mdMCI) if participants had z-scores < 1.5 in both the memory and nonmemory domain [43,44].

#### 2.4.3. Potential Confounders

Potential confounders were assessed at baseline.

Frailty Index—Out of Reference (FI-OutRef) is an abbreviated form of the FI-Lab frailty index and reflects overall organism dysfunction due to deficiencies in numerous organ systems [45]. FI-OutRef is calculated using 17 laboratory tests, which are often routinely assessed upon admission to the ED, including mean corpuscular volume, hemoglobin, mean corpuscular hemoglobin concentration, leukocytes, differential blood count, thrombocytes, C-reactive protein, creatinine, blood urea nitrogen, sodium, potassium, albumin, alanine aminotransferase, alkaline phosphatase, lactate dehydrogenase, bilirubin, and coagulation factors ll, Vll, and X (INR) [45]. FI-OutRef is calculated as the number of test results outside the age- and sex-specific reference interval divided by the total number of results for patients with ≥10 of 17 available results, multiplied by 17, to standardize FI-OutRef [45]. An FI-OutRef ≥5 is strongly associated with long-term mortality post-discharge [45].

Charlson Comorbidity Index (CCI) is an index based on weighted comorbidities used to measure disease burden with the ability to predict one-year mortality [46]. CCI was developed in 1987 [46] and later reevaluated and applied to hospital discharge data, using the International Classification of Diseases, 9th revision (ICD-9) [47,48,49]. In 2011, Quan et al. defined the CCI using the International Classification of Diseases, 10th revision (ICD-10) [50], and this algorithm was used to calculate CCI in this study, with higher CCI scores indicating a higher disease burden.

Medications marketed in Denmark potentially associated with cognitive decline were identified from the EU (7)-PIM list [51]. These medications were: Anxiolytics (N05BA), Hypnotics and sedatives (N05CF), Hydroxyzine (N05BB0), Amitriptyline (N06AA09), and Nortriptyline (N06AA10).

Further, age was considered a potential confounder.

#### 2.4.4. Descriptive Variables

The descriptive variables used in this study were collected using semi-structured interviews and measurements at baseline and 4-week follow-up. The descriptive variables included demographics (age, sex, smoking status, and education), anthropometry (body weight, Body Mass Index (BMI)), physical performance (hand grip strength (HGS), gait speed (GS)), social variables (living condition, need of assistance, health-related quality of life, and falls within the last year), and cognitive performance (Mini-Mental State Examination (MMSE)). Education was grouped as: primary school = 0–9 years of school; secondary school = 10–11 years of school, graduate or skilled; and higher education = short, medium-cycle, or long-cycle higher education. Body weight was measured on a scale (if not possible, self-reported weight was registered). BMI was calculated based on self-reported height and body weight. Physical performance was evaluated using hand grip strength measured by a hydraulic hand dynamometer (Digi-II, Saehan) [52] and habitual gait speed measured over a four-meter course from a standing start position [53]. Health-related quality of life was measured by subjective reporting on a Visual Analogue Scale (VAS) from 0 to 100 (100 being best health) as part of the EuroQol-5 dimensions-5 levels (EQ-5D-5L) (permission granted from EuroQol Research Foundation) [54,55]. The Mini-Mental State Examination (MMSE) is a screening test designed to evaluate cognitive impairment in older adults and was used for a descriptive purpose in this study. MMSE is used to screen for dementia in light to moderate form with varying cut-offs reported in the literature, but is not sensitive enough to identify MCI [40,56,57,58]. MMSE has a maximum total score of 30, with higher scores indicating better performance [59]. With a conservative approach, we used a cut-off of ≤24 for suspected dementia.

### 2.5. Data Sources

Data from FAM-Cph and STAND-Cph were entered directly in an electronic case report form (CRF) in the Research Electronic Data Capture (REDCap) (Vanderbilt University, Nashville, TN, USA) or written down on a paper CRF and double entered in REDCap to ensure data integrity and accuracy. To address data collection as a potential source of bias, clinical in-service training and supervision were provided by the same qualified person in both studies during data collection to enhance the quality of the measurements. Blood samples were routinely collected upon admission to the ED and the results from the blood samples were entered in Microsoft Excel before FI-OutRef was calculated (for further details please see Pedersen et al. [27] and Andersen et al. [29]).

### 2.6. Statistical Analyses

This study is an explorative study based on data from the STAND-Cph trial and the FAM-Cph trial. Descriptive data are presented as mean and standard deviation (SD) for continuous variables and as frequency (*n*) and percentage (%) for categorical variables. The prevalence of malnutrition and risk of malnutrition at baseline and 4-week follow-up based on MCI/noMCI are presented in a bar graph. Linear regression analyses were performed to examine the association between the independent variable for cognitive performance (MCI/noMCI) and the dependent variable for nutritional status (MNA-SF score). First, we calculated unadjusted estimates and confidence intervals (CI) for the difference in nutritional status (MNA-SF score) at baseline and 4-week follow-up between MCI and noMCI, and the difference in the change in MNA-SF score between baseline and 4-week follow-up between MCI and noMCI.

Additionally, adjusted models were fitted with age, and with age and one of either frailty, comorbidity, or medication, separately. Finally, a model including all possible confounders was fitted. In all the linear regression models, the independent variable (MCI/noMCI) and dependent variable (MNA-SF) had no missing values, but missing values for confounders were excluded from the analyses potentially giving a biased estimate. Further, we performed a sensitivity analysis that included participants who completed the baseline assessment and the 4-week follow-up assessment but who had not completed enough cognitive performance tests to be identified with MCI/noMCI (*n* = 54) to access possible selection bias. Participants with missing MCI/noMCI were sampled to MCI or noMCI randomly with a weighted probability such that the distribution of MCI/noMCI would match that of the non-missing participants. A total of 10.000 of these samples were generated and unadjusted analyses were repeated for each sample giving a distribution of estimates to be compared with the estimates based only on non-missing MCI/noMCI participants.

All analyses were performed as complete case analyses. The assumption of normal distribution was assessed by QQ-plots and statistical significance was determined at *p*-values < 0.05. All statistical analyses were performed using R Version 3.6.1 (R Foundation for Statistical Computing, Vienna, Austria).

## 3. Results

### 3.1. Participant Characteristics

A total of 205 participants completed the baseline assessment, and 143 participants completed both baseline and 4-week follow-up assessments, with an assessment of MCI being feasible in 89 (62%) participants (Figure 1).

MCI was present in 42/89 (47.2%) participants, and identification of MCI sub-types was feasible in 36/42 (85.7%) participants at the 4-week follow-up, with the following prevalence: aMCI (*n* = 3), naMCI (*n* = 3), and mdMCI (*n* = 30). Additionally, frailty measures were missing for two participants. Compared to participants with noMCI, participants with MCI had a higher age (mean 81.8 years vs. 77.1 years), poorer performance in GS (mean 0.7 m/s vs. 0.9 m/s), poorer performance in 30 s STS test (mean 7.0 vs. 10.0), and more use of assistance with regards to purchases (46.3% vs. 17.0%); dressing (26.8% vs. 2.1%, *p* = 0.001); and medication (43.9% vs. 4.3%) and laundry (48.8% vs. 12.8%) (Table 1). Further, suspected dementia was present in 11/42 (26.2%) of participants with MCI and 1/47 (2.1%) of participants with noMCI when using the conservative MMSE cut-off of ≤24.

At baseline, the prevalence of malnutrition was 10.6 % for participants without MCI and 31.0 % for participants with MCI. For risk of malnutrition, the corresponding prevalence was 51.1 % (noMCI) and 45.2 % (MCI) (Figure 2). At 4-week follow-up, the prevalence of malnutrition was 4.3 % (noMCI) and 28.6 % (MCI) and the prevalence of risk of malnutrition was 51.1 % (noMCI) and 52.4 % (MCI).

### 3.2. The Association between MCI and Nutritional Status at Baseline and 4-Week Follow-Up

MCI compared to noMCI was associated with poorer nutritional status on admission for the unadjusted model with an average difference of −1.29 points (CI: −2.30; −0.28) and the model adjusted for age, frailty, comorbidity, and medication with an average difference of −1.10 points (CI: −2.15; −0.06) (Table 2). All remaining models at baseline showed non-significant associations with estimates between −0.98 and −0.84. At 4-week follow up, MCI compared to noMCI was associated with poorer nutritional status with an average difference of −1.64 points (CI: −2.57; −0.70) for the unadjusted model; −1.29 points (CI: −2.25; −0.34) for the model adjusted for age; −1.43 points (CI: −2.39; −0.46) for the model adjusted for age and frailty; −1.29 points (CI: −2.25; −0.33) for the model adjusted for age and comorbidity; −1.29 points (CI: −2.27; −0.32) for the model adjusted for age and medication; and −1.45 points (CI: −2.44; −0.45) for the model adjusted for age, frailty, comorbidity, and medication (Table 2).

### 3.3. The Association between MCI and Change in Nutritional Status from Baseline to 4-Week Follow-Up

MCI was not associated with a change in nutritional status (MNA-SF score) between baseline and 4-week follow-up (change) in any models and the model estimates did not change substantially when adjusting for possible confounders. The change in MNA-SF from baseline to 4-week follow-up for participants with MCI and noMCI is visualized in Figure 3.

### 3.4. Sensitivity Analysis

Because 54/143 participants did not complete enough cognitive performance tests to be identified with MCI/noMCI, it was not possible to include these in the analyses (Table 2). Results from the sensitivity analysis including missing MCI/noMCI participants showed that 3.9% of sample estimates from baseline and 1.4% from 4-week follow-up had more extreme estimates than the estimates from the analyses (−1.29 and −1.64, respectively). Additionally, 22.8% of the performed sample estimates from baseline and 52.2% from 4-week follow-up had a sample estimate of less than −1 (equal to a clinically relevant difference) (Figure 4).

## 4. Discussion

### 4.1. Main Findings

To our knowledge, this is the first study that has investigated the association between MCI and nutritional status in acutely hospitalized older patients. Our study found that MCI is associated with nutritional status upon hospitalization and four weeks after discharge in acutely hospitalized older patients with the strongest association four weeks after discharge. No association was found between MCI and change in nutritional status from hospitalization to four weeks after discharge. Age explained some of the associations at hospitalization and the 4-week follow-up, and frailty, comorbidity, and medication did not change the interpretation of these results.

### 4.2. Results in the Context of Other Studies and Significant Findings

Having MCI might affect a person’s ability to receive and comply with dietary guidance because of impaired memory and/or processing speed [18]. Compliance with nutritional interventions has not yet been reported in acutely hospitalized older patients. However, a study on community-dwelling older adults found that cognitive impairment is a predictor of poor compliance with prescribed drug regimens [60]. Another study found that cognitive impairment is strongly correlated with treatment compliance and adherence in older hypertensive patients [61]. Therefore, MCI should be considered when designing interventions for acutely hospitalized older patients to avoid MCI influencing compliance with the intervention and thereby the effect of the intervention. The findings of this study imply that it is important to accommodate dietary counseling to the patient’s cognitive abilities. However, it is unknown whether such adaptations will improve the effect of and compliance with nutritional interventions in older hospitalized patients with MCI and further research is needed. The involvement of relatives and/or care providers might be necessary for successful prevention or treatment [10]. This is supported by our results (Table 1), where almost half of those with MCI already have an established contact to care providers, making the inclusion of care providers in dietary counseling feasible. For those without contact with any care provider, a larger effort is required. Hence, our results call for the development of new tools and strategies to accommodate limited cognitive abilities when preventing and treating malnutrition in this population. Additionally, when malnutrition or risk of malnutrition is identified in this population, care providers should consider systematic screening for MCI; however, simple tools for this purpose are warranted.

The prevalence of malnutrition or risk of malnutrition (68.5%) in this study is similar to the prevalence of 60–70% reported in the literature [3,5,62]. Additionally, we found a higher frequency of malnutrition and a lower frequency of normal nutrition in the MCI group compared to the noMCI group, which is similar to the findings in other studies [12,13,14]. Therefore, it is of great relevance to take the challenges of older adults with MCI into account when planning future nutritional studies on this population.

After adjusting for age, further individual adjustment for frailty, comorbidity, and medication affected the unadjusted MCI/noMCI estimates imperceptibly. This indicates that much of the potential confounding effect of frailty, comorbidity, and medication is confounded by age, as age expectedly will affect all three possible confounders. Therefore, individually, frailty, comorbidity, and medication explain only a small part of the association when age is accounted for. When accounting for all confounders simultaneously we see a larger change in the estimates suggesting that a larger part of the association can still be explained when accounting for all confounders. Further, different methods for the identification of frailty exist. In our study, we choose to assess frailty by using FI-OutRef, a cumulative deficits model of frailty. In an alternative approach, we could have assessed frailty using a phenotypic model as Fried’s criteria [63,64]. Therefore, we cannot rule out that frailty assessed differently would have explained some of the associations found between MCI and nutritional status. Although, we would still expect age to influence the estimates.

In the absence of acute illness (at four-week follow-up) the total MNA-SF score increases by two points in all patients by the simple fact that the acute illness item on the MNA-SF is scored differently. This might explain why we did not find an association between MCI and the change in nutritional status between admission and four weeks after discharge. The influence of these two points might hide any minor changes otherwise existing between the two groups because the two points constitute a significant share of the total change in the MNA-SF score. Therefore, the MNA-SF might not be sensitive enough to identify a significant change in nutritional status in this setting and might explain why we only found an association at baseline and 4-week follow-up. However, MNA-SF is the only screening tool that is validated in both settings, and therefore an obvious choice of screening tool, when assessing change in nutritional status in a transitional setting.

### 4.3. Strengths and Limitations

A strength of this study is that we used MCI identified at the 4-week follow-up for all analyses. Since the development of MCI from intact cognition rarely happens within a year [65] and the development of dementia from MCI seems to develop over 2–3 years [66], the likelihood of cognitive function changing significantly within four weeks is low. Therefore, the identification of MCI at the 4-week follow-up is reasonable to apply to all baseline analyses. Moreover, during hospitalization, the participants might have been under the influence of acute disease and medical treatment, which could temporarily affect their cognitive abilities if measured at baseline, and hereby overestimate the prevalence of MCI.

Our study has some limitations worth mentioning. With a difference in MNA-SF score ≥1 (chosen cut-off for a clinically relevant difference) between MCI and noMCI, our results support a clinically relevant association between MCI and nutritional status in two models at baseline and all models at 4-week follow-up. The cut-off of ≥1 was chosen based on previously described considerations, but whether it is a reasonable cut-off should be investigated. With four weeks between the baseline and follow-up assessments of MNA-SF, the assessment at baseline will inevitably influence the assessment at the 4-week follow-up since questions in the MNA-SF refer to the patients’ experiences within the past three months. Moreover, questions in the MNA-SF are seeking self-reported answers, potentially resulting in reporting bias. Additionally, hydration, anticholinergic burden, and infection status upon assessment of cognitive function were not measured, which might bias the result of the cognitive performance tests negatively [24,67]. Further, those who declined participation in FAM-Cph and STAND-Cph might be those with the poorest health and possibly the poorest cognition. Furthermore, given the reduced effect in most of the sensitivity estimates, results from the primary analysis might be overestimated. Analyses of MCI sub-types could provide additional valuable insights into the participants’ cognitive abilities since, e.g., amnestic impairment might affect the ability to follow nutritional recommendations differently than non-amnestic impairment. However, the limited sample size of 89 did not justify these analyses. Finally, we cannot rule out that those identified with MCI have a more severe degree of cognitive impairment, which is supported by the number of participants having an MMSE-score <24. However, we excluded those with diagnosed dementia and without the ability to comply cognitively in both FAM-Cph and STAND-Cph. In our study, 47% were identified with MCI. Among participants with MCI, 26% had an MMSE-score ≤24, compared to 2% among the cognitively intact participants. Considering that the prevalence of diagnosed dementia was 1.43% among adults ≥65 years in Denmark in 2018 [68], our results suggest a potentially higher prevalence of undiagnosed dementia, when having been acutely admitted. Undiagnosed dementia might lead to lost opportunities for treatment and considerations in regard to treatment and this may pose challenges for health care professionals [69]. These challenges may also apply to those with undetected MCI, which implies a possible need for dietary counseling to be more differentiated and better fitted with the patients’ compromised cognitive abilities when preventing and treating malnutrition in this population.

## 5. Conclusions

MCI is associated with poorer nutritional status at hospitalization and four weeks after discharge in acutely hospitalized older patients, with the strongest association four weeks after discharge. Our study did not find an association between MCI and change in nutritional status from hospitalization to four weeks after discharge. Age explained some of the association at hospitalization and four weeks after discharge, while frailty, comorbidity, and medication did not change the interpretation of these results. The association found between MCI and poorer nutritional status emphasizes the importance of potentially alternating dietary counseling to accommodate the patient’s cognitive abilities when preventing and treating malnutrition in acutely hospitalized older patients. Whether such alternations will improve the effect of and compliance with nutritional interventions in this population is unknown and further research on the effect is needed.

## Figures and Tables

**Figure 1 geriatrics-07-00095-f001:**
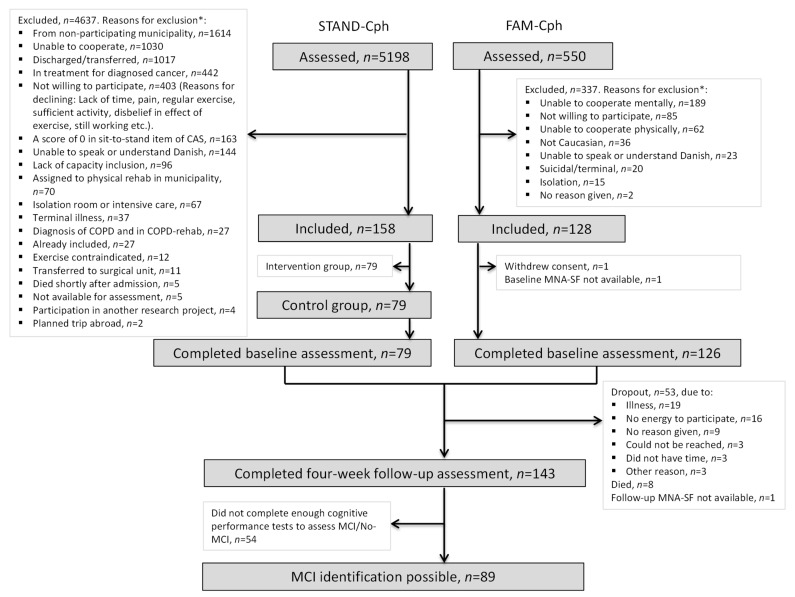
Flowchart. * Patients can have more than one reason for exclusion.

**Figure 2 geriatrics-07-00095-f002:**
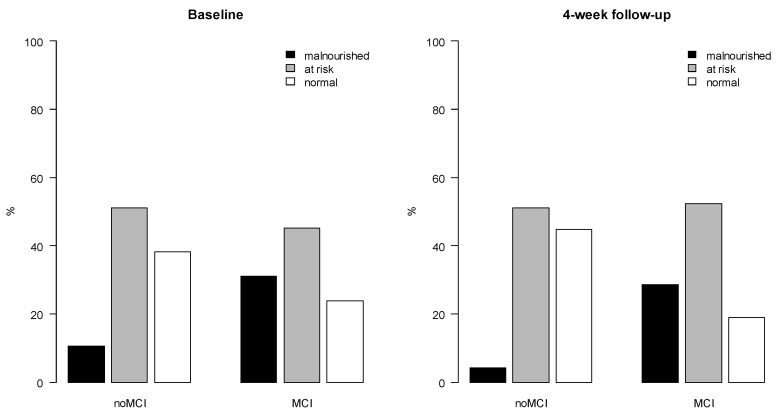
The prevalence of malnutrition and risk of malnutrition (assessed with MNA-SF) at baseline and 4-week follow-up for participants with mild cognitive impairment (MCI) and participants with intact cognition (noMCI). Notes to Figure 2: Malnourished = malnutrition (MNA-SF score 0–7); risk = risk of malnutrition (MNA-SF score of 8–11); normal = normal nutritional status (MNA-SF score 12–14); noMCI = intact cognition; MCI = mild cognitive impairment.

**Figure 3 geriatrics-07-00095-f003:**
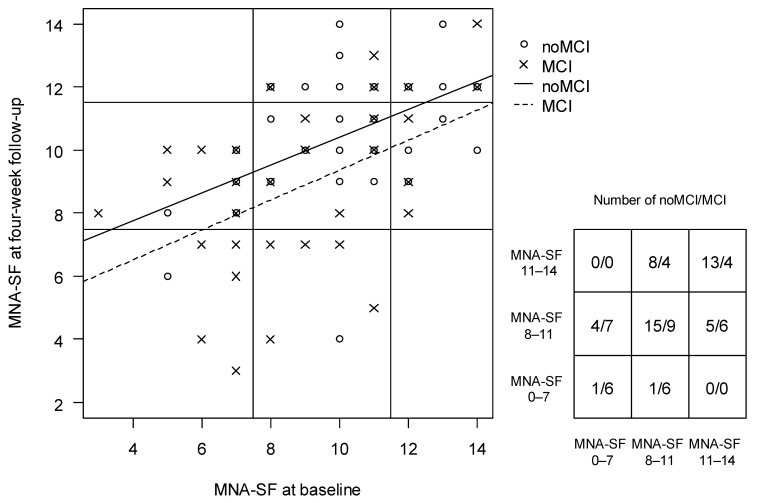
Nutritional status (MNA-SF score) at baseline and 4-week follow-up for participants with mild cognitive impairment (MCI) and participants with intact cognition (noMCI) with observation-matrix (MCI/noMCI), *n* = 89. MCI = Mild Cognitive Impairment; noMCI = Intact cognition, MNA-SF = Mini Nutritional Assessment—Short Form, MNA-SF score 0–7 = Malnutrition; MNA-SF score of 8–11 = risk of malnutrition; MNA-SF score 12–14 = normal nutritional status.

**Figure 4 geriatrics-07-00095-f004:**
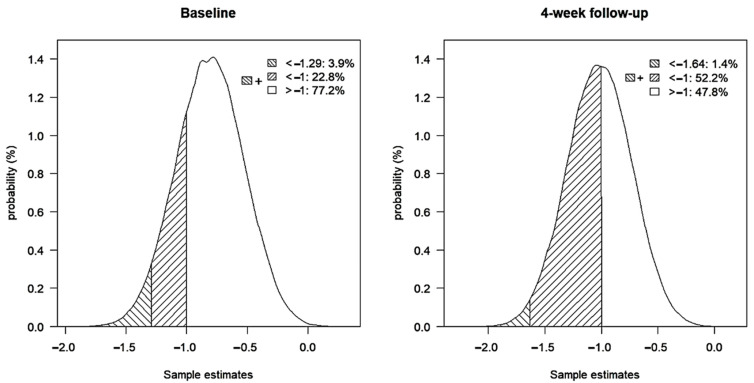
Distribution of sample estimates from sensitivity analysis.

**Table 1 geriatrics-07-00095-t001:** Characteristics of the study participants at 4-week follow-up (*n* = 89).

	MCI (*n* = 42)	*n*	noMCI (*n* = 47)	*n*
**Demographic Variables**				
Age	81.8 (9.1)	42	77.1 (6.0)	47
Sex, female, *n* (%)	20 (47.6)	42	34 (72.3)	47
Smoking, *n* (%) *		42		47
Yes	5 (11.9)		6 (12.8)	
Used to	20 (47.6)		19 (40.4)	
Never	17 (40.5)		22 (46.8)	
Education, *n* (%) *		41		47
Primary school	13 (31.7)		8 (17.0)	
Secondary education	20 (48.8)		29 (61.7)	
Higher education	8 (19.5)		10 (21.3)	
**Anthropometry**				
Body weight, kg	72.8 (23.8)	42	77.2 (18.5)	47
BMI ^a^	25.1 (6.3)	42	27.5 (5.4)	47
**Physical Performance**				
Max hand grip strength, kg	24.5 (12.1)	38	24.5 (9.8)	46
Gait speed, m/s	0.7 (0.3)	40	0.9 (0.3)	46
**Social Variables**				
Living alone, yes, *n* (%) *	29 (69.0)	42	31 (66.0)	47
Assistance, yes, *n* (%)				
Cleaning	29 (70.7)	41	25 (53.2)	47
Purchases	19 (46.3)	41	8 (17.0)	47
Dressing	11 (26.8)	41	1 (2.1)	47
Medication	18 (43.9)	41	2 (4.3)	47
Laundry	20 (48.8)	41	6 (12.8)	47
EQ-5D-5L ^b^, VAS ^c^	63.3 (18.3)		70.5 (18.6)	
**Cognitive Performance**				
MMSE-score ^d^ ≤ 24, *n* (%)	11 (26.2)	42	1 (2.1)	47

* Baseline measurement, ^a^ BMI = Body Mass Index; ^b^ EQ-5D-5L = EuroQol–5 Dimensions–5 Levels, ^c^ VAS = Visual Analogue Scale, ^d^ MMSE = Mini Mental State Examination.

**Table 2 geriatrics-07-00095-t002:** The difference in nutritional status (MNA-SF score) between participants with mild cognitive impairment compared to participants with intact cognition at baseline, 4-week follow-up, and the change between baseline, and 4-week follow-up (change) (*n* = 89).

MNA	Model Adjusted for	Beta (95 % CI)	*p*-Value
Baseline	Unadjusted	−1.29 (−2.30; −0.28)	0.013 *
	Age	−0.85 (−1.86; 0.17)	0.100
	Age, frailty **	−0.92 (−1.95; 0.10)	0.077
	Age, comorbidity	−0.84 (−1.87; −0.17)	0.102
	Age, medication	−0.98 (−2.00; 0.05)	0.061
	Age, frailty, comorbidity, medication **	−1.10 (−2.15; −0.06)	0.039 *
4 w fw	Unadjusted	−1.64 (−2.57; −0.70)	<0.001 *
	Age	−1.29 (−2.25; −0.34)	0.009 *
	Age, frailty **	−1.43 (−2.39; −0.46)	0.004 *
	Age, comorbidity	−1.29 (−2.25; −0.33)	0.009 *
	Age, medication	−1.29 (−2.27; −0.32)	0.010 *
	Age, frailty, comorbidity, medication **	−1.45 (−2.44; −0.45)	0.005 *
Change	Unadjusted	−0.34 (−1.32; 0.63)	0.486
	Age	−0.44 (−1.47; 0.58)	0.393
	Age, frailty **	−0.33 (−1.32; 0.67)	0.513
	Age, comorbidity	−0.44 (−1.47; 0.58)	0.393
	Age, medication	−0.24 (−1.22; 0.75)	0.632
	Age, frailty, comorbidity, medication **	−0.34 (−1.41; 0.72)	0.522

* Significant results, 4 w fw = 4-week follow-up, ** = models based on 87 participants due to missing frailty measures.

## Data Availability

The data presented in this study are available on request from the corresponding author. The data are not publicly available due to regulations set out by the Danish Data Protection Agency regarding data anonymization.

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
