# Peer review of "Mild Cognitive Impairment Is Associated with Poorer Nutritional Status on Hospital Admission and after Discharge in Acutely Hospitalized Older Patients"

_geriatrics, 2022, doi:10.3390/geriatrics7050095_

Round 1

Reviewer 1 Report

The paper studied the association between MCI and malnutrition among acutely hospitalized older adults. Based on the results, MCI is associated with poorer nutritional status at hospitalization and four weeks after discharge in acutely hospitalized older patients, with the strongest association four weeks after discharge. In comparison, this study did not find an association between MCI and change in nutritional status from hospitalization to four weeks after discharge. In my opinion, the results are important. However, some limitations are notable:

  1. Why did the authors didn’t consider sex as one of the confounders?

  2. The authors should provide more explication and interpretation of no association was found between MCI and change in nutritional status from hospitalization to four weeks after discharge, in relative to the significant findings.

  3. The authors mentioned in the introduction, “Since persons with MCI have a slight impairment of cognitive functions within memory and processing speed [19], they may lack the ability to understand and comply with dietary counselling.” 

Later, the authors concluded, “The association found between MCI and poorer nutritional status emphasizes the importance of potentially alternating dietary counselling to accommodate the patient’s cognitive abilities when preventing and treating malnutrition in acutely hospitalized older patients.” As we know those cognitively impaired may lack the ability to comply with the dietary instruction, can the authors elaborate on how to accommodate this? Provide more care, more supervision, or more follow-up visits?

  1. Page 6, line 279-281 has possible typos. “10.000 of these…” The sentence was not ended with a period.

Reviewer 2 Report

Overall, the article is of good academic quality. Therefore, there are only a few side notes. These are: 1) Figure 1 is very confusing and should be reduced to the essentials. 2) Table 1 is overloaded with content and should be shortened.  

Author Response

Reviewer #2
Comment 1: Figure 1 is very confusing and should be reduced to the essentials.
Reply: Thank you for this recommendation. We find the detailed information provided in ‘Figure 1’ necessary for transparency regarding the study participants, however, to make ‘Figure 1’ less confusing we have made visual changes.
Action taken: We have added point-format in ‘Figure 1’ where possible to make it less confusing (see the “Participant characteristics” section between lines 296-297 (TC)/293-294 (C)).

Comment 2: Table 1 is overloaded with content and should be shortened.
Reply: We agree that some of the descriptive variables in Table 1 are providing too much detail to the patient characteristics and that Table 1 should be shortened.
Action taken: We have removed five descriptive variables in ‘Table 1’ between lines 313-314 (TC)/305-306 (C) and in the ‘Method and Materials’ section between lines 227238 (TC)/226-235 (C).